# Deep Synoptic Monte Carlo Planning in Reconnaissance Blind Chess

**Gregory Clark**
ML Collective, Google
`gregoryclark@google.com`

## Abstract

This paper introduces deep synoptic Monte Carlo planning (DSMCP) for large imperfect information games. The algorithm constructs a belief state with an unweighted particle filter and plans via playouts that start at samples drawn from the belief state. The algorithm accounts for uncertainty by performing inference on "synopses," a novel stochastic abstraction of information states. DSMCP is the basis of the program `Penumbra`, which won the official 2020 reconnaissance blind chess competition versus 33 other programs. This paper also evaluates algorithm variants that incorporate caution, paranoia, and a novel bandit algorithm. Furthermore, it audits the synopsis features used in `Penumbra` with per-bit saliency statistics.

## 1   Introduction

Choosing a Nash equilibrium strategy is rational when the opponent is able to identify and exploit suboptimal behavior [Bowling and Veloso, 2001]. However, not all opponents are so responsive, and computing a Nash equilibrium is intractable for many games. This paper presents deep synoptic Monte Carlo planning (DSMCP), an algorithm for large imperfect information games that seeks a best-response strategy rather than a Nash equilibrium strategy.

When opponents use fixed policies, an imperfect information game may be viewed as a partially observable Markov decision process (POMDP) with the opponents as part of the environment. DSMCP treats playing against specific opponents as related offline reinforcement learning (RL) problems and exploits predictability. Importantly, the structure of having opponents with imperfect information is preserved in order to account for their uncertainty.

DSMCP uses sampling to break the "curse of dimensionality" [Pineau et al., 2006] in three ways: sampling possible histories with a particle filter, sampling possible futures with upper confidence bound tree search (UCT) [Kocsis and Szepesvári, 2006], and sampling possible world states within each information state uniformly. It represents information states with a generally-applicable stochastic abstraction technique that produces a "synopsis" from sampled world states. This paper assesses DSMCP on reconnaissance blind chess (RBC), a large imperfect information chess variant.

## 2   Background

Significant progress has been made in recent years in both perfect and imperfect information settings. For example, using deep neural networks to guide UCT has enabled monumental achievements in abstract strategy games as well as computer games [Silver et al., 2016, 2017a,b, Schrittwieser et al., 2019, Wu, 2020, Tomašev et al., 2020]. This work employs deep learning in a similar fashion.

Recent advancements in imperfect information games are also remarkable. Several programs have reached superhuman performance in Poker [Moravčík et al., 2017, Brown and Sandholm, 2018, 2019, Brown et al., 2020]. In particular, ReBeL [Brown et al., 2020] combines RL and search by converting

35th Conference on Neural Information Processing Systems (NeurIPS 2021).

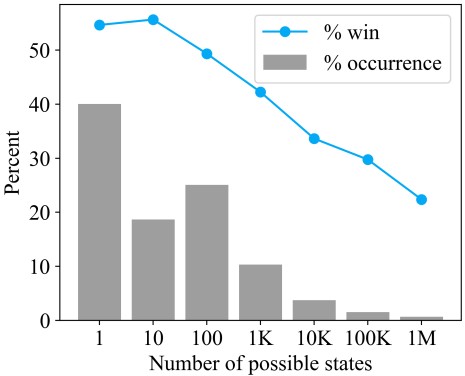

Figure 1: Within 98k historical reconnaissance blind chess (RBC) games between non-random players, the win percentage tends to decrease as the number of possible states increases. The games were replayed while tracking up to one million states. Each bucket is labeled with an inclusive upper bound. The median of the maximum number of possible states encountered during a game is 4,869.

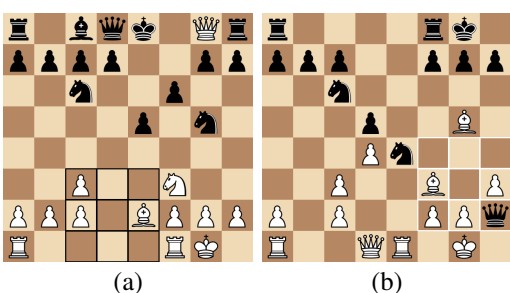

(a)                    (b)

Figure 2: Playing RBC well requires balancing risks and rewards. (a) On the left, `Penumbra` moved the white queen to `g8`. After sensing at `d2`, Black could infer that the white queen occupied one of 25 squares. That uncertainty allowed the white queen to survive and capture the black king on the next turn. (b) On the right, `Penumbra` moved the black queen to `h2`. In this case, the opponent detected and captured the black queen. The games are available online at `https://rbc.jhuapl.edu/games/120174` and `https://rbc.jhuapl.edu/games/124718`.

imperfect information games into continuous state space perfect information games with public belief states as nodes. This approach is powerful, but it relies on public knowledge and fails to scale to games with hidden actions and substantial private information, such as RBC.

Information set search [Parker et al., 2006, 2010] is a limited-depth algorithm for imperfect information games that operates on information states according to a minimax rule. This algorithm was designed for and evaluated on Kriegspiel chess, which is comparable to RBC.

Partially observable Monte Carlo planning (POMCP) [Silver and Veness, 2010] achieves optimal policies for POMDPs by tracking approximate belief states with an unweighted particle filter and planning with a variant of UCT on a search tree of histories. In practice, POMCP can suffer from particle depletion, requiring a domain-specific workaround. This work combines an unweighted particle filter with a novel information state abstraction technique which increases sample quality and supports deep learning.

Smooth UCT [Heinrich and Silver, 2015] and information set Monte Carlo tree search (ISMCTS) [Cowling et al., 2012] may be viewed as multi-agent versions of POMCP. These two algorithms for playing extensive-form games build search trees (for each player) of information states. These two algorithms and DSMCP all perform playouts from determinized states that are accurate from the current player's perspective, effectively granting the opponent extra information. Still, Smooth UCT approached a Nash equilibrium by incorporating a stochastic bandit algorithm into its tree search. DSMCP uses a similar bandit algorithm that mixes in a learned policy during early node visits.

While adapting perfect information algorithms has performed surprisingly well in some imperfect information settings [Long et al., 2010], the theoretical guarantees of variants of counterfactual regret minimization (CFR) [Neller and Lanctot, 2013, Brown et al., 2018] are enticing. Online outcome sampling (OOS) [Lisy et al., 2015] extends Monte Carlo counterfactual regret minimization (MCCFR) [Lanctot et al., 2009] by building its search tree incrementally and targeting playouts to relevant parts of the tree. OOS draws samples from the beginning of the game. MCCFR and OOS are theoretically guaranteed to converge to a Nash equilibrium strategy. Specifically, CFR-based algorithms produce mixed strategies while DSMCP relies on incidental stochasticity.

Neural fictitious self-play (NFSP) [Heinrich and Silver, 2016] is an RL algorithm for training two neural networks for imperfect information games. Experiments with NFSP employed compact observations embeddings of information states. DSMCP offers a generic technique for embedding information states in large games. Dual sequential Monte Carlo (DualSMC) [Wang et al., 2019] estimates belief states and plans in a continuous domain via sequential Monte Carlo with heuristics.

# 3 Reconnaissance blind chess

Reconnaissance blind chess (RBC) [Newman et al., 2016, Markowitz et al., 2018, Gardner et al., 2020] is a chess variant that incorporates uncertainty about the placement of the opposing pieces along with a private sensing mechanism. As shown in Figure 1, RBC players are often faced with thousands of possible game states, and reducing uncertainty increases the odds of winning.

**Game rules**   Pieces move in the same way in RBC as in chess. Players cannot directly observe the movement of the opposing pieces. However, at the beginning of each turn, players may view the ground truth of any $3\times3$ patch of the board. The information gained from the sensing action remains private to that player. Players are also informed of the location of all captures, but not the identity of capturing pieces. When a requested move is illegal, the move is substituted with the closest legal move and the player is notified of the substitution. For example, in Figure 2 (a), if Black attempted to move the rook from `h8` to `f8`, the rook would capture the queen on `g8` and stop there instead. Players are always able to track the placement of their own pieces. Capturing the opposing king wins the game, and players are not notified about check. Passing and moving into check are legal.

**Official competition**   This paper introduces the program `Penumbra`, the winner of the official 2020 RBC rating competition. In total, 34 programs competed to achieve the highest rating by playing public games. Ratings were computed with *BayesElo* [Coulom, 2008], and playing at least 100 games was required to be eligible to win. Figure 2 shows ground truth positions from the tournament in which `Penumbra` voluntarily put its queen in danger. Players were paired randomly, but the opponent's identity was provided at the start of each game which allowed catering strategies for specific opponents. However, opponents were free to change their strategies at any point, so attempting to exploit others could backfire. Nonetheless, `Penumbra` sought to model and counter predictable opponents rather than focusing on finding a Nash equilibrium.

**Other RBC programs**   RBC programs have employed a variety of algorithms [Gardner et al., 2020] including Q-learning [Mnih et al., 2013], counterfactual regret minimization (CFR) [Zinkevich et al., 2008], online outcome sampling (OOS) [Lisy et al., 2015], and the Stockfish chess engine [Romstad et al., 2020]. Another strong RBC program [Highley et al., 2020, Blowitski and Highley, 2021] maintains a probability distribution for each piece. Most RBC programs select sense actions and move actions in separate ways while DSMCP unifies all action selection. Savelyev [2020] also applied UCT to RBC and modeled the root belief state with a distribution over 10,000 tracked positions. Input to a neural network consisted of the most-likely 100 positions, and storing a single training example required 3.5MB on average which was large enough to hinder training. This work represents training examples with compact synopses which are less than 1kB without compression.

# 4 Terminology

Consider the two-player extensive-form game with agents $\mathcal{P} = \{\text{self, opponent}\}$, actions $\mathcal{A}$, "ground truth" world states $\mathcal{X}$, and initial state $x_0 \in \mathcal{X}$. Each time an action is taken, each agent $p \in \mathcal{P}$ is given an observation $\mathbf{o}_p \in \mathcal{O}$ that matches ($\sim$) the possible world states from $p$'s perspective. For simplicity, assume the game has deterministic actions such that each $a \in \mathcal{A}$ is a function $a : X \to \mathcal{X}$ defined on a subset of world states $X \subset \mathcal{X}$. Define $\mathcal{A}_x$ as the set of actions available from $x \in \mathcal{X}$.

An information state (infostate) $s \in \mathcal{S}$ for agent $p$ consists of all observations $p$ has received so far.[1] Let $\mathcal{X}_s \subset \mathcal{X}$ be the set of all world states that are possible from $p$'s perspective from $s$. In general, $\mathcal{X}_s$ contains less information than $s$ since some (sensing) actions may not affect the world state. Define a collection of limited-size world state sets $\mathcal{L} = \{L \subset \mathcal{X}_s : s \in \mathcal{S}, |L| \le \ell\}$, given a constant $\ell$.

Let $\rho : \mathcal{X} \to \mathcal{P}$ indicate the agent to act in each world state. Assume that $\mathcal{A}_x = \mathcal{A}_y$ and $\rho(x) = \rho(y)$ for all $x, y \in \mathcal{X}_s$ and $s \in \mathcal{S}$. Then extend the definitions of actions available $\mathcal{A}_*$ and agent to act $\rho$ over sets of world states and over infostates in the natural way. A policy $\pi(a|s)$ is a distribution over actions given an infostate. A belief state $B(h)$ is a distribution over action histories. Creating a belief state from an infostate requires assumptions about the opponent's action policy $\tau(a|s)$. Let

---

[1]An infostate is equivalent to an information set, which is the set of all possible action histories from $p$'s perspective [Osborne and Rubinstein, 1994].

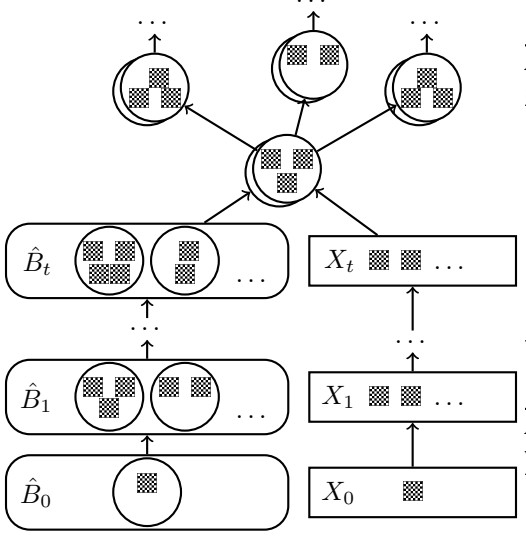

Figure 3: DSMCP approximates infostates with size-limited sets of possible states (circles). It tracks all possible states $X_t$ for each turn from its own perspective and constructs belief states $\hat{B}_t$ with approximate infostates from the opponent's perspective. At the root of each playout, the initial approximate infostate for the opponent is sampled from $\hat{B}_t$, and the initial approximate infostate for itself is a random subset of $X_t$.

---

**Algorithm 1** `Bandit` – Action selection with a stochastic multi-armed bandit

> **Given:** $c > 0, m \geq 0$
> **Input:** policy $\pi$, visit counts $\vec{n}$, value totals $\vec{q}$
> $n \leftarrow \sum_a \vec{n}_a$
> **if** $e^{-mn} > \texttt{Uniform}([0,1])$ and $\neg$root **then**
> $\quad$ **return** $a \leftarrow$ an action selected by policy $\pi$
> **else**
> $\quad$ **return** $a \leftarrow \operatorname{argmax}_a \left( \frac{\vec{q}_a}{\vec{n}_a} + c\pi_a \sqrt{\frac{\ln n}{\vec{n}_a}} \right)$

---

**Algorithm 2** `DrawSample` – Sample selection with rejection

> **Given:** $k, \ell \in \mathbb{Z}^+$, synopsis function $\sigma$,
> $\qquad\quad$ policy $\hat{\tau}$
> **Input:** previous belief distribution $\hat{B}$, possible
> $\qquad\quad$ states $X$, visit counts $\mathbf{N}$, value totals $\mathbf{Q}$
> **for** $0$ **to** $k$ **do**
> $\quad J \leftarrow$ random sample from $\hat{B}$
> $\quad a \leftarrow \texttt{Bandit}(\hat{\tau}(\sigma(J)), \mathbf{N}_J, \mathbf{Q}_J)$
> $\quad I \leftarrow \{a(x) : x \in J\}$
> $\quad I \leftarrow$ random $\ell$ states from $I$ **if** $|I| > \ell$
> $\quad$ **return** $I$ **if** $I \cap X \neq \varnothing$
> **return** $\{$ random state from $X$ $\}$

---

$\mathcal{R}_p : \mathcal{X} \to \mathbb{R}$ map terminal states to the reward for player $p$. Then $(\mathcal{S}, \mathcal{A}, \mathcal{R}_{\text{self}}, \tau, s_0)$ is a POMDP, where the opponent's policy $\tau$ provides environment state transitions and $s_0$ is the initial infostate. In the rest of this paper, the word "state" refers to a world state unless otherwise specified.

## 5 Deep synoptic Monte Carlo planning

Effective planning algorithms for imperfect information games must model agents' choice of actions based on (belief states derived from) infostates, not on world states themselves. Deep synoptic Monte Carlo planning (DSMCP) approximates infostates with size-limited sets of possible world states in $\mathcal{L}$. It uses those approximations to construct a belief state and as UCT nodes [Kocsis and Szepesvári, 2006]. Figure 3 provides a high-level visualization of the algorithm.

A bandit algorithm chooses an action during each node visit, as described in Algorithm 1. This bandit algorithm is similar to Smooth UCB [Heinrich and Silver, 2015] in that they both introduce stochasticity by mixing in a secondary policy. Smooth UCB empirically approached a Nash equilibrium utilizing the average policy according to action visits at each node. DSMCP mixes in a neural network's policy ($\pi$) instead. The constant $c$ controls the level of exploration, and $m$ controls how the policy $\pi$ is mixed into the bandit algorithm. For example, taking $m = 0$ always selects actions directly with $\pi$ without considering visit counts, and taking $m = \infty$ never mixes in $\pi$. As in Silver et al. [2016], $\pi$ provides per-action exploration values which guide the tree search.

Approximate belief states are constructed as subsets $\hat{B} \subset \mathcal{L}$, where each $L \in \hat{B}$ is a set of possible world-states from the opponent's perspective. This "second order" representation of belief states allows DSMCP to account for the opponent's uncertainty. Infostates sampled with rejection (Algorithm 2) are used as the "particles" in a particle filter which models successive belief states. Sampling is guided by a neural network policy ($\hat{\tau}$) based on the identity of the opponent. To counter particle deprivation, if $k$ consecutive candidate samples are rejected as incompatible with the possible world states, then a singleton sample consisting of a randomly-chosen possible state is selected instead.

| **Algorithm 3** `ChooseAction` – UCT playouts | **Algorithm 4** `PlayGame` – DSMCP |
|---|---|

**Algorithm 3** `ChooseAction` – UCT playouts

**Given:** $b, d, \ell, n_{vl}, z \in \mathbb{Z}^+$, synopsis func. $\sigma$,
policy $\pi$, opp. policy $\hat{\tau}$, value func. $\nu$
**Input:** belief distribution $\hat{B}$, possible states $X$,
visit counts $\mathbf{N}$, value totals $\mathbf{Q}$
$\bar{\pi} \leftarrow$ average of $\pi \circ \sigma$ over $b$ samples from $\hat{B}$
**while** time is left **do**
    $J \leftarrow$ random sample from $\hat{B}$
    $I \leftarrow$ random $\ell$ states in $X$ including one in $J$
    $a_0 \leftarrow \texttt{Bandit}(\bar{\pi}, \mathbf{N}_{\text{root}}, \mathbf{Q}_{\text{root}})$
    **for** $t = 0$ **to** $d$ **do**
        $x_t \leftarrow$ the one state in $I \cap J$
        $\mathbf{o} \leftarrow$ observations when $a_t$ is played on $x_t$
        $I \leftarrow \{a(x) : x \in I, a \in \mathcal{A}_I, \mathbf{o}_{\text{self}} \sim a(x)\}$
        $J \leftarrow \{a(x) : x \in J, a \in \mathcal{A}_J, \mathbf{o}_{\text{opp}} \sim a(x)\}$
        $I \leftarrow$ random $\ell$ states in $I$ including $a_t(x_t)$
        $J \leftarrow$ random $\ell$ states in $J$ including $a_t(x_t)$
        $K_t \leftarrow I$ and $\mu \leftarrow \pi$ **if** $I$ is to act
        $K_t \leftarrow J$ and $\mu \leftarrow \hat{\tau}$ **if** $J$ is to act
        $d \mathrel{+}= 1$ **if** $\mathbf{N}_{K_t, a_t} > z$
        **for** $i = 0$ **to** $t$ **do**
            $\mathbf{N}_{K_i, a_i} \mathrel{+}= n_{vl}$
        $a_{t+1} \leftarrow \texttt{Bandit}(\mu(\sigma(K_t)), \mathbf{N}_{K_t}, \mathbf{Q}_{K_t})$
        $q \leftarrow \nu(\sigma(K_t))$
        **for** $i = 0$ **to** $t$ **do**
            $\mathbf{Q}_{K_i, a_i} \mathrel{+}= q$ **if** $\rho(K_t) = \rho(K_i)$ **else** $-q$
            $\mathbf{N}_{K_i, a_i} \mathrel{+}= 1 - n_{vl}$
**return** $a \leftarrow \text{argmax}_a \mathbf{Q}_{\text{root}}$

**Algorithm 4** `PlayGame` – DSMCP

**Given:** $n_{\text{particles}} \in \mathbb{Z}^+$
$\mathbf{N}_{*,*} \leftarrow \mathbf{0}$ // Visits $\forall$ (sample, action) $\in \mathcal{L} \times \mathcal{A}$
$\mathbf{Q}_{*,*} \leftarrow \mathbf{0}$ // Values $\forall$ (sample, action) $\in \mathcal{L} \times \mathcal{A}$
$X_0 \leftarrow \{x_0\}$
$\hat{B}_0 \leftarrow \{X_0\}$
**while** the game is not over **do**
    $t \leftarrow$ current turn
    $\mathbf{o} \leftarrow$ current observation
    *// Track all possible world states*
    $X_t \leftarrow \{a(x) : x \in X_{t-1}, a \in \mathcal{A}_x, \mathbf{o}_{\text{self}} \sim a(x)\}$
    *// Filter belief states with the new information*
    **for** $i = t - 1$ **to** $0$ **do**
        $X_i \leftarrow \{x \in X_i : \exists a \in \mathcal{A}_x$
                    such that $a(x) \in X_{i+1}\}$
        $\hat{B}_i \leftarrow \{I \in \hat{B}_i : I \cap X_i \neq \varnothing\}$
    *// Repopulate belief states with new particles*
    **while** opponent to act or $|\hat{B}_t| < n_{\text{particles}}$ **do**
        $i \leftarrow$ smallest $i > 0$ s.t. $|\hat{B}_i| < n_{\text{particles}}$
        $I \leftarrow \texttt{DrawSample}(\hat{B}_{i-1}, X_i, \mathbf{N}, \mathbf{Q})$
        $\hat{B}_i \leftarrow \hat{B}_i \cup \{I\}$
    **if** self to act **then**
        $a \leftarrow \texttt{ChooseAction}(\hat{B}_t, X_t, \mathbf{N}, \mathbf{Q})$
        Perform action $a$

The tree search, described in Algorithm 3, tracks an approximate infostate for each player while simulating playouts. Playouts are also guided by policy ($\pi$ and $\hat{\tau}$) and value ($\nu$) estimations from a neural network. A synopsis function $\sigma$ creates a fixed-size summary of each node as input for the network. The constant $b$ is the batch size for inference, $d$ is the search depth, $\ell$ is the size of approximate infostates, $n_{vl}$ is the virtual loss weight, and $z$ is a threshold for increasing search depth.

Algorithm 4 describes how to play an entire game, tracking all possible world states. Approximate belief states ($\hat{B}_t$) are constructed for each past turn by tracking $n_{\text{particles}}$ elements of $\mathcal{L}$ (from the opponent's point of view) with an unweighted particle filter. Each time the agent receives a new observation, all of the (past) particles that are inconsistent with the observation are filtered out and replenished, starting with the oldest belief states.

## 5.1 Synopsis

One of the contributions of this paper is the methodology used to approximate and encode infostates. Games that consist of a fixed number of turns, such as poker, admit a naturally-compact infostate representation based on observations [Heinrich and Silver, 2015, 2016]. However, perfect representations are not always practical. Game abstractions are often used to reduce computation and memory requirements. For example, imperfect recall is an effective abstraction when past actions are unnecessary for understanding the present situation [Waugh et al., 2009, Lanctot et al., 2012].

DSMCP employs a stochastic abstraction which represents infostates with sets of world states and then subsamples to a manageable cardinality ($\ell$). Finally, a permutation-invariant synopsis function $\sigma$ produces fixed-size summaries of the approximate infostates which are used for inference. An alternative is to run inference on "determinized" world states individually and then somehow aggregate the results. However, such aggregation can easily lead to strategy fusion [Frank et al., 1998]. Other alternatives include evaluating states with a recurrent network [Rumelhart et al., 1986] one-at-a-time or using a permutation-invariant architecture [Zaheer et al., 2017, Wagstaff et al., 2019].

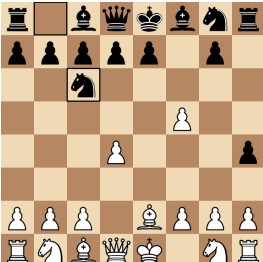

| # | White | Black |
|---|---|---|
| 1 | g6 : e2e4 | e3 : h7h5 |
| 2 | g7 : d2d4 | f2 : f7f5 |
| 3 | g6 : e4f5 | e4 : h5h4 |
| 4 | d7 : f1e2 | g4 : b8c6 |
| 5 | g7 : e2h5 | g4 : h8h5 |
| 6 | b7 : d1h5 | g5 : g7g6 |
| 7 | e7 : h5e8 | g6 : d7d6 |
| 8 | g6 : g6e8 | |

Figure 4: `Penumbra` played as White in this short game. In the position shown on the left, Black just moved a knight from `b8` to `c6`. From White's perspective, the black pieces could be placed in 238 different ways. Figure 5 shows a set of synopsis bitboards for this information state.

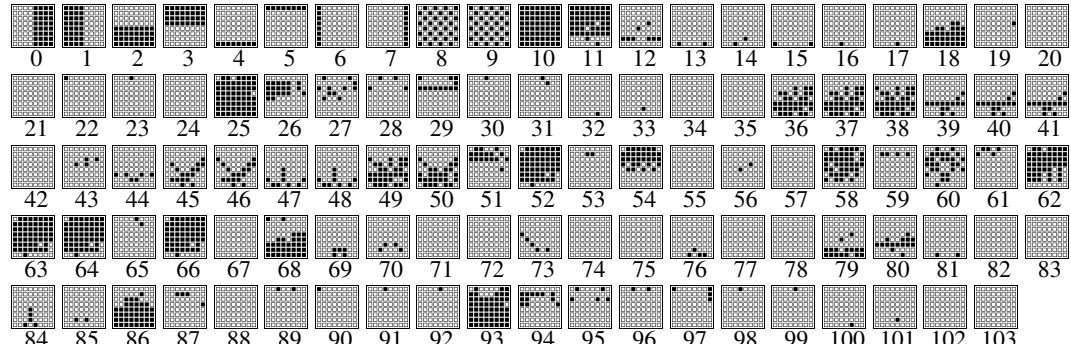

Figure 5: This set of synopsis bitboards was used as input to the neural network before White's sense on turn 5 of the game in Figure 4. The synopsis contains 104 bitboards. Each bitboard encodes 64 binary features of the possible state set that the synopsis summarizes. For example, bitboard #26 contains the possible locations of opposing pawns, and bitboard #27 contains the possible locations of opposing knights. An attentive reader may notice that the black pawn on `h4` is missing from bitboard #26, which is due to subsampling to $\ell = 128$ states before computing the bitboards. In this case, the true state was missing from the set of states used to create the synopsis. The features in each synopsis are only approximations of the infostates that they represent. The first 10 bitboards are constants, which provide information that is difficult for convolutions to construct otherwise [Liu et al., 2018].

Given functions $g_i : \mathcal{X} \to \{0, 1\}$ for $i = 0, \ldots, F$ that map states to binary features, define the $i^{\text{th}}$ component of a synopsis function $\sigma : \mathcal{L} \to \{0, 1\}^F$ as

$$\sigma_i(X) = g_i(x_0) *_i g_i(x_1) *_i \cdots *_i g_i(x_\ell) \tag{1}$$

where $X = \{x_0, x_1, \ldots, x_\ell\}$ and $*_i$ is either the logical AND ($\wedge$) or the logical OR ($\vee$) operation. For example, if $g_i$ encodes whether an opposing knight can move to the `d7` square of a chess board and $*_i = \wedge$, then $\sigma_i$ indicates that a knight can definitely move to `d7`. Figure 4 shows an example game, and Figure 5 shows an example output of `Penumbra`'s synopsis function, which consists of 104 bitboard feature planes each with 64 binary features. The appendix describes each feature plane.

## 5.2  Network architecture

`Penumbra` uses a residual neural network [He et al., 2016] as shown in Figure 6. The network contains 14 headsets, designed to model specific opponents and regularize each other as they are trained on different slices of data [Zhang et al., 2020]. Each headset contains 5 heads: a policy head, a value head, two heads for predicting winning and losing within the next 5 actions, and a head for guessing the number of pieces of each type in the ground truth world state. The `Top` policy head and the `All` value head are used for planning as $\pi$ and $\nu$, respectively. The other heads (including the `SoonWin`, `SoonLose`, and `PieceCount` heads) provide auxiliary tasks for further regularization [Wu, 2020, Fifty et al., 2020]. While playing against an opponent that is "recognized" (when a headset was trained on data from only that opponent), the policy head ($\hat{\tau}$) of the corresponding headset is used for

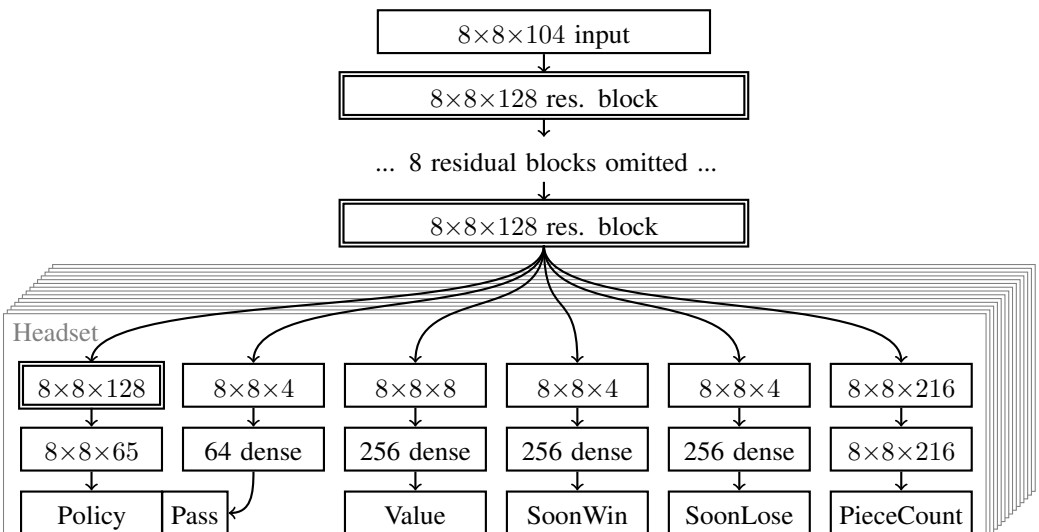

Figure 6: `Penumbra`'s network contains a shared tower with 10 residual blocks and 14 headsets. Each headset contains 5 heads for a total of 70 output heads. The residual blocks are shown with a double border, and they each contain two 3×3 convolutional layers and batch normalization. All of the convolutional layers in the headsets are 1×1 convolutions with the exception of the one residual block for each policy head. Each headset was trained on a separate subset of the data, as described in Table 1. The policy head provides logits for both sense and move actions.

the opponent's moves while progressing the particle filter (Algorithm 2) and while constructing the UCT tree (Algorithm 3). When the opponent is unrecognized, the `Top` policy head is used by default.

## 5.3 Training procedure

The network was trained on historical[2] game data as described by Table 1. The reported accuracies are averages over 5 training runs. The `All` headset was trained on all games, the `Top` headset was trained on games from the highest-rated players, the `Human` headset was trained on all games played by humans, and each of the other 11 headsets were trained to mimic specific opponents.

10% of the games were used as validation data based on game filename hashes. Training examples were extracted from games multiple times since reducing possible state sets to $\ell$ states is non-deterministic. A single step of vanilla stochastic gradient descent was applied to one headset at a time, alternating between headsets according to their training weights. See the appendix for hyperparameter settings and accuracy cross tables. Training and evaluation were run on four RTX 2080 Ti GPUs.

## 5.4 Implementation details

`Penumbra` plays RBC with DSMCP along with RBC-specific extensions. First, sense actions that are dominated by other sense actions are pruned from consideration. Second, `Penumbra` can detect some forced wins in the sense phase, the move phase, and during the opponent's turn. This static analysis is applied at the root and to playouts; playouts are terminated as soon as a player could win, avoiding unnecessary neural network inference. The static analysis was also used to clean training games in which the losing player had sufficient information to find a forced win.

Piece placements are represented with bitboards [Browne, 2014], and the tree of approximate infostates is implemented with a hash table. Zobrist hashing [Zobrist, 1990] maintains hashes of piece placements incrementally. Hash table collisions are resolved by overwriting older entries. The tree was not implemented until after the competition, so fixed-depth playouts were used instead ($m = 0$). Inference is done in batches of 256 during both training and online planning. The time used per

---

[2]The games were downloaded from `rbmc.jhuapl.edu` in June, 2019 and `rbc.jhuapl.edu` in August, 2020. Additionally, 5,000 games were played locally by `StockyInference`.

Table 1: Summary of headset training data and resulting validation accuracies

| Name | # Games | # Train Actions | # Validation Actions | Multiplicity | Training Weight | Stop Gradient | % Top-1 Accuracy | % Top-5 Accuracy | % Winner Accuracy |
|---|---|---|---|---|---|---|---|---|---|
| All | 85.6k | 16.8M | 1.8M | 4 | 15 | No | 33.6 | 62.0 | 74.8 |
| Top | 49.5k | 16.0M | 1.6M | ∼9.6 | 13 | No | 43.1 | 73.4 | 82.4 |
| StrangeFish | 13.3k | 8.1M | 0.8M | ∼20.7 | 10 | No | 45.9 | 74.3 | 86.6 |
| LaSalle | 3.7k | 4.3M | 0.4M | 32 | 4 | No | 36.4 | 68.2 | 69.7 |
| Dyn.Entropy | 7.8k | 7.8M | 0.8M | 32 | 7 | No | 50.0 | 76.7 | 78.9 |
| Oracle | 20.3k | 17.6M | 1.9M | 32 | 9 | No | 49.3 | 77.4 | 81.4 |
| StockyInfe. | 10.7k | 13.5M | 1.4M | 16 | 9 | No | 45.2 | 73.6 | 68.3 |
| Marmot | 10.2k | 3.9M | 0.4M | 16 | 8 | No | 24.6 | 55.4 | 80.8 |
| Genetic | 5.6k | 2.3M | 0.2M | 16 | 8 | No | 40.4 | 70.5 | 77.9 |
| Zugzwang | 10.9k | 3.1M | 0.3M | ∼12.0 | 5 | No | 47.0 | 71.6 | 80.1 |
| Trout | 18.0k | 5.1M | 0.5M | 16 | 5 | No | 41.8 | 63.4 | 79.8 |
| Human | 6.3k | 1.5M | 0.2M | 16 | 2 | No | 24.9 | 51.9 | 73.3 |
| Attacker | 15.3k | 1.7M | 0.2M | 16 | 4 | No | 45.1 | 61.9 | 80.1 |
| Random | 17.0k | 0.6M | 76k | 2 | 1 | Yes | 4.5 | 20.7 | 94.7 |

action is approximately proportional to the time remaining. The program processes approximately 4,000 nodes per second, and it plays randomly when the number of possible states exceeds 9 million.

## 6 Experiments

This section presents the results of playing games between `Penumbra` and several publicly available RBC baselines [Gardner et al., 2020, Bernardoni, 2020]. Each variant of `Penumbra` in Table 2 played 1000 games against each baseline, and each variant in Table 3 and Table 4 played 250 games against each baseline. Games with errors were ignored and replayed. The Elo ratings and 95% confidence intervals were computed with *BayesElo* [Coulom, 2008] and are all compatible. The scale was anchored with `StockyInference` at 1500 based on its rating during the competition.

Table 2 gives ratings of the baselines and five versions of `Penumbra`. `PenumbraCache` relied solely on the network policy for action selection in playouts ($m = 0$), `PenumbraTree` built a UCT search tree ($m = \infty$), and `PenumbraMixture` mixed in the network policy during early node visits ($m = 1$). The mixed strategy performed the best. `PenumbraNetwork` selected actions based on the network policy without performing any playouts. `PenumbraSimple` is the same as `PenumbraMixture` with the static analysis described in Section 5.4 disabled. `PenumbraNetwork` and `PenumbraSimple` serve as ablation studies; removing the search algorithm is detrimental while the effect of removing the static analysis is not statistically significant. Unexpectedly, `Penumbra` played the strongest against `StockyInference` when that program was unrecognized. So, in this case, modeling the opponent with a stronger policy outperformed modeling it more accurately.

Two algorithmic modifications that give the opponent an artificial advantage during planning were investigated. Table 3 reports the results of a grid search over "cautious" and "paranoid" variants of DSMCP. The caution parameter $\kappa$ specifies the percentage of playouts that use $\ell = 4$ for the opponent instead of the higher default limit. Since each approximate infostate is guaranteed to contain the correct ground truth in playouts, reducing $\ell$ for the opponent gives the opponent higher-quality information, allowing the opponent to counter risky play more easily in the constructed UCT tree.

The paranoia parameter augments the exploration values in Algorithm 1 to incorporate the minimum value seen during the current playout. With paranoia $\phi$, actions are selected according to

$$\underset{a}{\mathrm{argmax}} \left( (1 - \phi)\frac{\vec{q}_a}{\vec{n}_a} + \phi\vec{m}_a + c\pi_a\sqrt{\frac{\ln n}{\vec{n}_a}} \right) \qquad (2)$$

where $\vec{m}$ contains the minimum value observed for each action. This is akin to the notion of paranoia studied by Parker et al. [2006, 2010].

Table 2: Bot Elo scores

| Bot | Recognized as | **Elo score** |
|---|---|---|
| PenumbraMixture | unrecognized | $1747 \pm 11$ |
| PenumbraSimple | unrecognized | $1739 \pm 10$ |
| PenumbraCache | unrecognized | $1727 \pm 10$ |
| PenumbraTree | unrecognized | $1641 \pm 9$ |
| StockyInference | Trout | $1610 \pm 7$ |
| StockyInference | StrangeFi. | $1528 \pm 8$ |
| StockyInference | Genetic | $1512 \pm 8$ |
| StockyInference | StockyInf. | $1500 \pm 8$ |
| StockyInference | unrecognized | $1474 \pm 8$ |
| PenumbraNetwork | unrecognized | $1376 \pm 9$ |
| AggressiveTree | unrecognized | $1134 \pm 15$ |
| FullMonte | unrecognized | $1028 \pm 20$ |
| Trout | Trout | $1005 \pm 22$ |
| Trout | unrecognized | $997 \pm 22$ |

Table 3: Caution and paranoia grid search results

| Paranoia | Caution 0% | 10% | 20% | 30% |
|---|---|---|---|---|
| 0% | $1711 \pm 19$ | $1714 \pm 19$ | $1707 \pm 19$ | $1702 \pm 19$ |
| 10% | $1711 \pm 19$ | $1705 \pm 19$ | $1726 \pm 19$ | $1695 \pm 18$ |
| 20% | $1688 \pm 18$ | $1700 \pm 19$ | $1688 \pm 18$ | $1670 \pm 18$ |
| 30% | $1691 \pm 18$ | $1683 \pm 18$ | $1681 \pm 18$ | $1666 \pm 18$ |

Table 4: Exploration strategy grid search results

| | Exploration ratio $c$ | | |
|---|---|---|---|
| | 1 | 2 | 4 |
| UCB1 | $1698 \pm 19$ | $1686 \pm 18$ | $1696 \pm 18$ |
| aVoP | $1696 \pm 18$ | $1680 \pm 18$ | $1695 \pm 18$ |

Table 4 shows the results of a grid search over exploration constants and two bandit algorithms. UCB1 [Kuleshov and Precup, 2014] (with policy priors), which is used on the last line of Algorithm 1, is compared with "a variant of PUCT" (aVoP) [Silver et al., 2016, Tian et al., 2019, Lee et al., 2019], another popular bandit algorithm. This experiment used $\kappa = 20\%$ and $\phi = 20\%$. Figure 7 show that `Penumbra`'s value head accounts for the uncertainty of the underlying infostate.

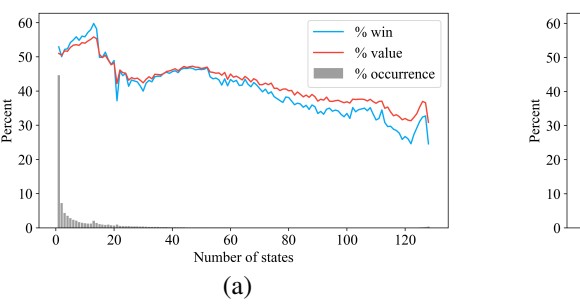
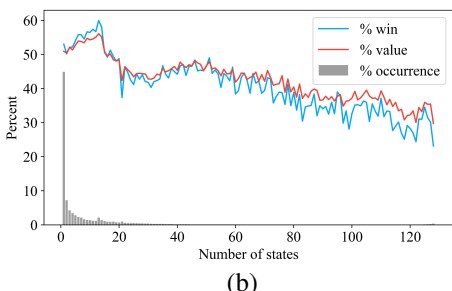

(a)      (b)

Figure 7: The mean historical win percentage and the mean network value assigned to (a) train and (b) validation synopses tend to decrease as the number of world states given to $\sigma$ increases.

# 7 Per-bit saliency

Saliency methods may be able to identify which of the synopsis feature planes are most important and which are least important. Gradients only provide local information, and some saliency methods fail basic sanity checks [Adebayo et al., 2018]. Higher quality saliency information may be surfaced by integrating gradients over gradually-varied inputs [Sundararajan et al., 2017, Kapishnikov et al., 2019] and by smoothing gradients locally [Smilkov et al., 2017]. Those saliency methods are not directly applicable to discrete inputs such as the synopses used in this work. So, this paper introduces a saliency method that aggregates gradient information across two separate dimensions: training examples and iterations. Per-batch saliency (PBS) averages the absolute value of gradients over random batches of test examples throughout training. Similarly, per-bit saliency (PbS) averages the absolute value of gradients over bits (with specific values) within batches of test examples throughout training. Gradients were taken both with respect to the loss and with respect to the action policy.

Figure 8 shows saliency information for input synopsis features used by `Penumbra`. In order to validate that these saliency statistics are meaningful, the model was retrained 104 times, once with each feature removed [Hooker et al., 2018]. Higher saliency is slightly correlated with decreased performance when a feature is removed. The correlation coefficient to the average change in accuracy is $-0.208$ for loss-PBS, and $-0.206$ for action-PbS. Explanations for the low correlation include noise in the training process and the presence of closely-related features. Ultimately, the contribution of a feature during training is distinct from how well the model can do without that feature. Since

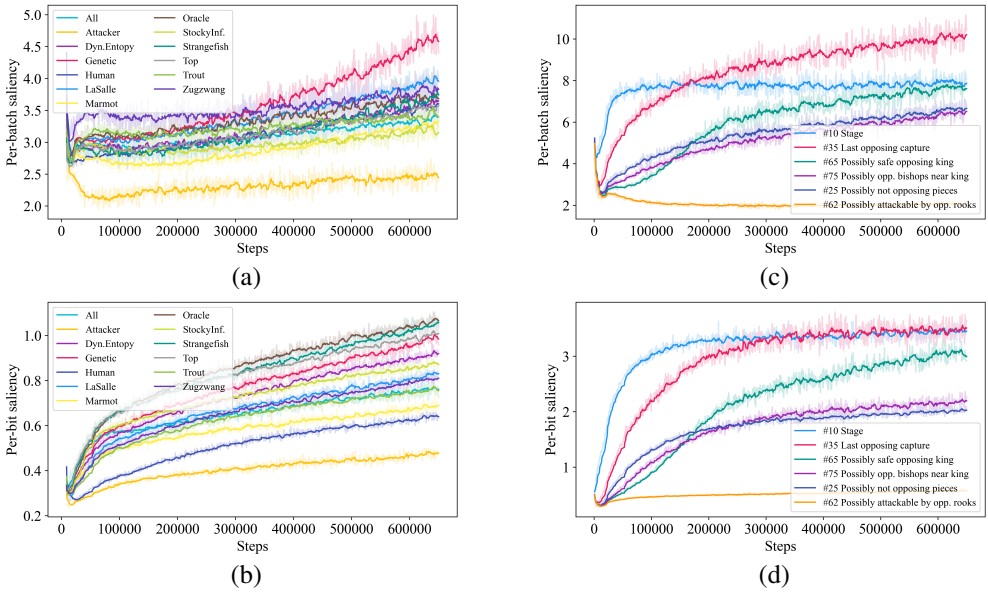

Figure 8: (a) The loss per-batch saliency (PBS) and (b) the action per-bit saliency (PbS) are taken on test examples during training. These graphs show the saliency of feature plane #8, dark squares, for each headset in one training run. The large gradients with respect to the loss suggest that the `Genetic` headset has overfit. (c) The loss PBS and (d) the action PbS provide insight about which synopsis features are most useful. The top-five most-salient feature planes and the least-salient feature plane for the `Top` headset from one training run are shown.

some features are near-duplicates of others, removing one may simply increase dependence on another. Still, features with high saliency — such as the current stage (sense or move) and the location of the last capture — are likely to be the most important, and features with low saliency may be considered for removal. The appendix includes saliency statistics for each feature plane.

# 8 Discussion

**Broader impact** DSMCP is more broadly applicable than some prior algorithms for imperfect information games, which are intractable in settings with large infostates and small amounts of shared knowledge [Brown et al., 2020]. RBC and the related game Kriegspiel were motivated by uncertainty in warfare [Newman et al., 2016]. While playing board games is not dangerous in itself, algorithms that account for uncertainty may become effective and consequential in the real world. In particular, since it focuses on exploiting weaknesses of other agents, DSMCP could be applied in harmful ways.

**Future work** Planning in imperfect information games is an active area of research [Russell and Norvig, 2020], and RBC is a promising testing ground for such research. `Penumbra` would likely benefit from further hyperparameter tuning and potentially alternative corralled bandit algorithms [Arora et al., 2020]. Modeling an opponent poorly could be catastrophic; algorithmic adjustments may lead to more-robust best-response strategies [Ponsen et al., 2011]. How much is lost by collapsing infostates with synopses is unclear and deserves further investigation. Finally, the "bitter lesson" of machine learning [Sutton, 2019] suggests that a learned synopsis function may perform better.

# Acknowledgements

Thanks to the Johns Hopkins University Applied Physics Laboratory for inventing such an intriguing game and for hosting RBC competitions. Thanks to Ryan Gardner for valuable correspondence. Thanks to Rosanne Liu, Joel Veness, Marc Lanctot, Zhe Zhao, and Zach Nussbaum for providing feedback on early drafts. Thanks to William Bernardoni for open sourcing high-quality baseline bots. Thanks to Solidmind for the song "Penumbra", which is an excellent soundtrack for programming.

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
