# Appendix for Deep Synoptic Monte Carlo Planning in Reconnaissance Blind Chess

**Gregory Clark**
ML Collective, Google
gregoryclark@google.com

## A   Appendix

Table 1 lists the training hyperparameters and runtime hyperparameters used by `PenumbraMixture`. Table 2, Table 3, and Table 4 provide top-1 action, top-5 action, and winner accuracies, respectively, between each headset in the neural network. Figure 1 shows game length distributions for each headset.

The synopsis features were hand-designed. Many of them are natural given the rules of chess. Some of them are near duplicates of each other. Table 5 and Table 6 jointly provide brief descriptions of each synopsis feature plane. These tables also include saliency estimates averaged over five runs. The penultimate column orders the synopsis features by their per-bit saliency based on action gradients, and the final column reports the average difference of the policy head accuracies when the model was retrained without each feature.

Table 1: Hyperparameters used by `PenumbraMixture`

| Symbol | Parameter | Value |
|---|---|---|
| $b$ | Batch size | 256 |
| $c$ | Exploration constant | 2 |
| $d_{\text{sense}}$ | Search depth for sense actions | 6 |
| $d_{\text{move}}$ | Search depth for move actions | 12 |
| $F$ | # of binary synopsis features | $8{\times}8{\times}104$ |
| $k$ | Rejection sampling persistence | 512 |
| $\ell$ | Limited state set size | 128 |
| $m$ | Bandit mixing constant | 1 |
| $n_{\text{particles}}$ | # of samples to track | 4096 |
| $n_{\text{vl}}$ | Virtual loss | 1 |
| $n_{\text{batches}}$ | Total minibatches of training | 650000 |
| $n_{\text{width}}$ | Network width; # features per layer | 128 |
| $n_{\text{depth}}$ | Network depth; # residual blocks | 10 |
| $z$ | Depth increase threshold | $\infty$ |
| $\kappa$ | Caution | 0 |
| $\phi$ | Paranoia | 0 |
| $\epsilon$ | Learning rate | 0.0005 |

### A.1   2019 NeurIPS competition

`Penumbra` was originally created to compete in the 2019 reconnaissance blind chess competition hosted by the Conference on Neural Information Processing Systems (NeurIPS). However, it performed very poorly in that competition, winning fewer games than the random bot.

The program and underlying algorithm presented in this paper are largely the same as the originals. The main differences are that some hyperparameters were adjusted, the neural network was retrained with more data, and a key bug in the playout code was fixed. Instead of choosing actions according to

35th Conference on Neural Information Processing Systems (NeurIPS 2021).

the policy from the neural network, the playout code erroneously always selected the last legal action. Giving the program a `break` made a huge difference.

## A.2 Comparison with Kriegspiel

A comparison between RBC and Kriegspiel chess [Ciancarini and Favini, 2009, Parker et al., 2010, Richards, 2012] may be worthwhile. Kriegspiel chess also introduces uncertainty about the opposing pieces but lacks an explicit sensing mechanism. Instead, information is gathered solely from captures, check notifications, and illegal move attempts. In Kriegspiel, illegal moves are rejected and the player is allowed to choose a new move with their increased information about the board state, which entangles the positional and informational aspects of the game. In contrast, sensing in RBC gives players direct control over the amount and character of the information they possess.

Another significant difference comes from the mechanics related to check. Capturing pieces and putting the opposing king into check have benefits in both games: capturing pieces leads to a material advantage, and check often precedes checkmate. In Kriegspiel, however, both capturing and giving check also provide the opponent with information. In RBC, while capturing does give the opponent information, putting their king into check does not, which makes sneak attacks more viable.

## A.3 Games played

The games that were played in order to produce the main results are available for download from `https://github.com/w-hat/penumbra`.

## References

Paolo Ciancarini and Gian Piero Favini. Monte carlo tree search techniques in the game of kriegspiel. In *IJCAI*, pages 474–479, 2009. URL `http://ijcai.org/Proceedings/09/Papers/086.pdf`.

Austin Parker, Dana S. Nau, and V. S. Subrahmanian. Paranoia versus overconfidence in imperfect-information games. In Rina Dechter, Hector Geffner, and Joseph Y. Halpern, editors, *Heuristics, Probabilities, and Causality: A Tribute to Judea Pearl*, chapter 5, pages 63–87. College Publications, 2010.

Mark D. Richards. Reasoning and decisions in partially observable games. 2012.

Table 2: Top-1 action accuracy across headsets.

| | Dataset | | | | | | | | | | | | | |
|---|---|---|---|---|---|---|---|---|---|---|---|---|---|---|
| Headset | All | Top | StrangeFish | LaSalle | Dyn.Entropy | Oracle | StockyInfe. | Marmot | Genetic | Zugzwang | Trout | Human | Attacker | Random |
| All | **33.6** | 41.5 | 40.5 | 32.3 | 43.7 | 44.1 | 41.5 | 20.3 | 31.9 | 37.1 | 36.6 | 21.9 | 41.4 | 3.8 |
| Top | 30.5 | **43.1** | 44.5 | 32.3 | 43.9 | 44.4 | 40.3 | 19.4 | 31.6 | 22.4 | 26.0 | 18.5 | 15.6 | 3.3 |
| StrangeFish | 27.3 | 40.5 | **45.9** | 30.3 | 36.9 | 38.8 | 33.2 | 17.9 | 27.7 | 21.0 | 23.6 | 17.5 | 14.6 | 3.3 |
| LaSalle | 26.0 | 34.4 | 34.6 | **36.4** | 34.1 | 34.5 | 33.2 | 17.2 | 24.6 | 22.6 | 26.1 | 15.5 | 11.2 | 3.4 |
| Dyn.Entropy | 27.9 | 37.9 | 35.2 | 27.5 | **50.0** | 40.1 | 37.4 | 18.0 | 32.6 | 18.3 | 22.6 | 16.9 | 13.8 | 3.3 |
| Oracle | 28.8 | 38.4 | 36.3 | 29.1 | 41.2 | **49.3** | 35.1 | 17.2 | 29.5 | 19.9 | 26.6 | 17.0 | 11.9 | 3.4 |
| StockyInfe. | 28.8 | 38.1 | 33.9 | 30.7 | 42.9 | 38.6 | **45.2** | 17.9 | 29.3 | 18.4 | 23.1 | 16.7 | 11.6 | 3.3 |
| Marmot | 22.4 | 29.4 | 28.6 | 24.0 | 31.4 | 29.3 | 29.7 | **24.6** | 25.0 | 16.4 | 15.5 | 15.4 | 11.1 | 3.4 |
| Genetic | 24.0 | 32.5 | 30.3 | 24.1 | 39.8 | 35.0 | 32.3 | 16.9 | **40.4** | 15.1 | 15.7 | 15.1 | 7.8 | 3.4 |
| Zugzwang | 20.5 | 21.8 | 23.2 | 20.5 | 20.4 | 23.5 | 17.3 | 11.7 | 12.3 | **47.0** | 34.6 | 14.0 | 10.9 | 3.2 |
| Trout | 22.8 | 25.1 | 26.0 | 24.0 | 23.0 | 27.8 | 21.3 | 12.5 | 14.4 | 36.1 | **41.8** | 14.9 | 14.7 | 3.7 |
| Human | 23.8 | 30.1 | 30.6 | 24.9 | 31.4 | 30.1 | 28.4 | 16.8 | 24.1 | 24.7 | 24.5 | **24.9** | 12.5 | 3.3 |
| Attacker | 10.6 | 11.7 | 11.0 | 9.8 | 11.4 | 11.6 | 12.4 | 8.6 | 8.8 | 9.2 | 9.0 | 6.7 | **45.1** | 4.4 |
| Random | 14.0 | 16.7 | 16.2 | 14.0 | 16.5 | 17.8 | 16.8 | 9.4 | 11.4 | 15.7 | 16.5 | 10.4 | 10.4 | **4.5** |

Table 3: Top-5 action accuracy across headsets.

| | Dataset | | | | | | | | | | | | | |
|---|---|---|---|---|---|---|---|---|---|---|---|---|---|---|
| Headset | All | Top | StrangeFish | LaSalle | Dyn.Entropy | Oracle | StockyInfe. | Marmot | Genetic | Zugzwang | Trout | Human | Attacker | Random |
| All | **62.0** | 72.3 | 71.0 | 66.6 | 74.9 | 75.8 | 72.5 | 51.3 | 65.3 | 65.5 | 61.8 | 48.1 | 59.1 | 18.2 |
| Top | 58.8 | **73.4** | 73.4 | 66.4 | 75.5 | 76.4 | 72.0 | 50.1 | 64.7 | 48.5 | 52.2 | 45.4 | 35.3 | 16.3 |
| StrangeFish | 56.8 | 72.2 | **74.3** | 64.7 | 72.6 | 74.2 | 67.5 | 48.5 | 62.1 | 46.5 | 50.1 | 43.8 | 41.4 | 15.7 |
| LaSalle | 56.8 | 68.8 | 68.4 | **68.2** | 69.8 | 70.6 | 68.3 | 48.7 | 58.9 | 51.5 | 56.6 | 43.6 | 30.8 | 16.8 |
| Dyn.Entropy | 55.5 | 69.1 | 66.9 | 60.9 | **76.7** | 72.3 | 69.6 | 47.2 | 64.4 | 38.4 | 47.2 | 42.7 | 41.4 | 16.6 |
| Oracle | 56.7 | 70.4 | 68.9 | 61.9 | 74.1 | **77.4** | 69.1 | 45.6 | 63.4 | 43.9 | 50.6 | 43.1 | 34.4 | 16.4 |
| StockyInfe. | 57.0 | 70.0 | 67.3 | 64.7 | 74.0 | 71.3 | **73.6** | 49.3 | 63.3 | 43.3 | 50.2 | 44.0 | 29.8 | 17.0 |
| Marmot | 53.4 | 65.0 | 64.0 | 58.7 | 68.6 | 66.1 | 65.2 | **55.4** | 59.3 | 43.0 | 46.2 | 42.6 | 32.2 | 16.2 |
| Genetic | 52.6 | 65.8 | 64.2 | 58.2 | 71.3 | 69.7 | 65.8 | 45.4 | **70.5** | 37.4 | 41.6 | 40.9 | 27.0 | 16.2 |
| Zugzwang | 42.8 | 45.0 | 45.4 | 46.0 | 42.1 | 47.5 | 42.4 | 32.5 | 32.8 | **71.6** | 57.9 | 35.3 | 29.0 | 15.4 |
| Trout | 49.3 | 54.5 | 54.4 | 53.9 | 53.7 | 57.2 | 52.5 | 38.4 | 42.2 | 62.9 | **63.4** | 38.7 | 40.9 | 18.1 |
| Human | 53.5 | 62.9 | 62.4 | 58.3 | 64.7 | 64.6 | 62.5 | 45.9 | 55.7 | 53.7 | 53.4 | **51.9** | 33.4 | 16.3 |
| Attacker | 35.2 | 39.5 | 38.8 | 34.7 | 40.8 | 40.2 | 39.7 | 31.1 | 33.8 | 33.0 | 32.4 | 28.4 | **61.9** | 20.0 |
| Random | 39.7 | 45.6 | 44.5 | 41.2 | 46.4 | 47.9 | 46.0 | 31.8 | 37.7 | 41.3 | 42.1 | 31.5 | 30.3 | **20.7** |

Table 4: Winner accuracy across headsets.

| Headset \ Dataset | All | Top | StrangeFish | LaSalle | Dyn.Entropy | Oracle | StockyInfe. | Marmot | Genetic | Zugzwang | Trout | Human | Attacker | Random |
|---|---|---|---|---|---|---|---|---|---|---|---|---|---|---|
| All | **74.8** | 73.4 | 76.6 | 68.4 | 74.6 | 76.3 | 67.8 | 79.3 | 74.1 | 76.7 | 76.7 | 72.1 | 79.6 | 91.3 |
| Top | 63.1 | **82.4** | **86.7** | 68.0 | 76.0 | 80.6 | 66.3 | 65.6 | 73.2 | 49.1 | 55.1 | 52.7 | 47.7 | 29.9 |
| StrangeFish | 64.1 | 82.1 | 86.6 | 67.5 | 76.2 | 80.6 | 65.2 | 65.8 | 72.7 | 50.1 | 55.4 | 55.8 | 60.9 | 37.3 |
| LaSalle | 69.9 | 77.3 | 80.7 | **69.7** | 76.0 | 78.8 | 67.1 | 73.5 | 75.9 | 65.1 | 68.3 | 62.4 | 71.9 | 60.7 |
| Dyn.Entropy | 67.8 | 80.4 | 85.0 | 69.4 | **78.9** | 80.8 | 67.0 | 71.7 | 75.6 | 58.5 | 61.1 | 61.5 | 65.0 | 47.3 |
| Oracle | 66.0 | 82.0 | 86.4 | 69.4 | 77.3 | **81.4** | 66.9 | 69.2 | 75.3 | 53.5 | 58.2 | 57.9 | 59.5 | 39.7 |
| StockyInfe. | 71.3 | 78.5 | 82.5 | 69.3 | 77.2 | 79.5 | **68.3** | 75.3 | 76.8 | 64.0 | 65.8 | 66.9 | 72.6 | 68.9 |
| Marmot | 70.6 | 67.7 | 70.1 | 64.5 | 70.9 | 72.3 | 65.7 | **80.8** | 72.9 | 73.9 | 73.4 | 69.2 | 77.0 | 72.6 |
| Genetic | 67.1 | 80.0 | 84.1 | 68.6 | 77.1 | 80.3 | 67.1 | 71.5 | **77.9** | 56.5 | 60.8 | 60.9 | 62.1 | 44.3 |
| Zugzwang | 68.3 | 54.6 | 54.0 | 59.7 | 61.4 | 60.0 | 61.7 | 76.1 | 64.4 | **80.1** | 78.8 | 72.5 | 78.5 | 88.9 |
| Trout | 69.7 | 58.6 | 59.1 | 60.9 | 63.6 | 64.4 | 63.4 | 77.9 | 69.0 | 77.9 | **79.8** | 72.4 | 78.6 | 87.3 |
| Human | 71.1 | 64.8 | 66.0 | 63.8 | 67.7 | 68.0 | 65.1 | 76.6 | 70.4 | 74.6 | 76.2 | **73.3** | 77.7 | 90.1 |
| Attacker | 63.7 | 47.1 | 46.1 | 55.6 | 53.8 | 50.9 | 56.5 | 71.4 | 59.5 | 75.0 | 73.4 | 71.5 | **80.1** | 92.7 |
| Random | 51.0 | 25.5 | 20.9 | 43.4 | 34.8 | 28.5 | 46.5 | 54.9 | 38.2 | 65.7 | 56.6 | 62.5 | 69.5 | **94.7** |

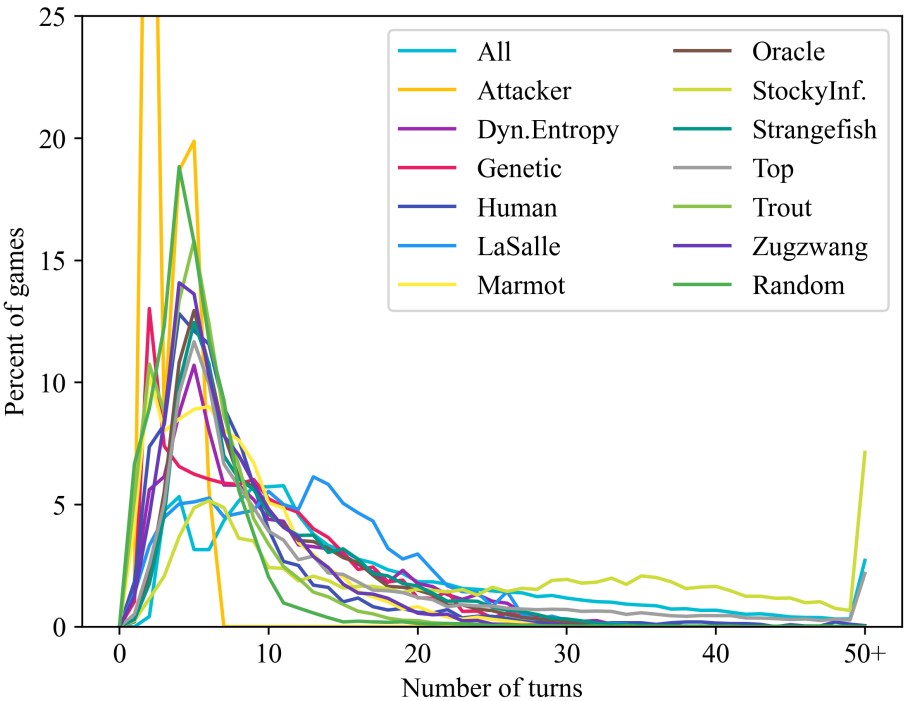

Figure 1: The historical game length distributions are shown for the data used to train each of the headsets. On average, the games from `Attacker` were the shortest, and the games from `StockyInference` where the longest.

Table 5: Synopsis feature descriptions, saliency estimates, and ablation study results.

| # | Description | Loss PBS | Action PbS | Action PbS 0 | Action PbS 1 | Action PbS # | Ablation % Acc. Diff. |
|---|---|---|---|---|---|---|---|
| 0 | East side (constant) | 3.24 | 0.86 | 0.80 | 0.91 | 34 | 0.27 |
| 1 | West side (constant) | 3.12 | 0.82 | 0.84 | 0.81 | 43 | 0.00 |
| 2 | South side (constant) | 3.24 | 0.86 | 0.78 | 0.94 | 33 | -0.08 |
| 3 | North side (constant) | 3.22 | 0.82 | 0.89 | 0.75 | 44 | 0.27 |
| 4 | Rank 1 (constant) | 3.15 | 0.80 | 0.81 | 0.76 | 50 | -0.03 |
| 5 | Rank 8 (constant) | 3.12 | 0.82 | 0.85 | 0.61 | 42 | -0.07 |
| 6 | A-file (constant) | 3.04 | 0.78 | 0.81 | 0.53 | 59 | 0.18 |
| 7 | H-file (constant) | 3.08 | 0.80 | 0.83 | 0.58 | 51 | 0.15 |
| 8 | Dark squares (constant) | 3.08 | 0.82 | 0.81 | 0.82 | 45 | 0.21 |
| 9 | Light squares (constant) | 3.03 | 0.78 | 0.77 | 0.78 | 61 | 0.01 |
| 10 | Stage (move or sense) | 7.80 | 3.14 | 2.82 | 3.45 | 0 | -0.19 |
| 11 | Not own piece | 5.40 | 1.43 | 2.66 | 1.13 | 8 | -0.29 |
| 12 | Own pawns | 4.16 | 1.14 | 1.07 | 1.73 | 14 | 0.01 |
| 13 | Own knights | 3.68 | 0.93 | 0.91 | 1.63 | 22 | 0.09 |
| 14 | Own bishops | 3.46 | 0.89 | 0.87 | 1.63 | 27 | 0.02 |
| 15 | Own rooks | 3.67 | 0.94 | 0.93 | 1.12 | 21 | 0.03 |
| 16 | Own queens | 3.28 | 0.87 | 0.85 | 2.24 | 32 | 0.06 |
| 17 | Own king | 3.14 | 0.79 | 0.79 | 0.88 | 52 | 0.10 |
| 18 | Definitely not opposing pieces | 3.85 | 1.14 | 1.03 | 1.21 | 13 | -0.10 |
| 19 | Definitely opposing pawns | 3.49 | 1.01 | 1.02 | 0.73 | 17 | -0.08 |
| 20 | Definitely opposing knights | 3.30 | 0.93 | 0.93 | 0.61 | 23 | -0.05 |
| 21 | Definitely opposing bishops | 3.21 | 0.88 | 0.88 | 0.59 | 29 | -0.02 |
| 22 | Definitely opposing rooks | 3.04 | 0.81 | 0.82 | 0.38 | 47 | 0.02 |
| 23 | Definitely opposing queens | 3.15 | 0.85 | 0.85 | 0.60 | 35 | -0.10 |
| 24 | Definitely opposing king | 3.60 | 0.92 | 0.91 | 2.27 | 26 | 0.04 |
| 25 | Possibly not opposing pieces | 5.22 | 1.54 | 1.34 | 1.56 | 5 | -0.04 |
| 26 | Possibly opposing pawns | 3.50 | 0.92 | 0.92 | 0.93 | 24 | 0.06 |
| 27 | Possibly opposing knights | 2.97 | 0.77 | 0.77 | 0.81 | 67 | 0.07 |
| 28 | Possibly opposing bishops | 2.95 | 0.75 | 0.74 | 0.89 | 70 | 0.09 |
| 29 | Possibly opposing rooks | 3.01 | 0.75 | 0.76 | 0.63 | 69 | -0.18 |
| 30 | Possibly opposing queens | 3.05 | 0.78 | 0.77 | 1.05 | 57 | -0.07 |
| 31 | Possibly opposing kings | 4.86 | 1.48 | 1.43 | 2.64 | 7 | -0.04 |
| 32 | Last from | 2.77 | 0.72 | 0.72 | 0.83 | 76 | -0.11 |
| 33 | Last to | 3.28 | 0.96 | 0.96 | 1.40 | 19 | 0.02 |
| 34 | Last own capture | 3.10 | 0.83 | 0.83 | 1.17 | 40 | 0.07 |
| 35 | Last opposing capture | 8.04 | 2.83 | 2.82 | 6.51 | 1 | -0.08 |
| 36 | Definitely attackable | 2.72 | 0.70 | 0.62 | 0.78 | 84 | -0.06 |
| 37 | Definitely attackable somehow | 2.73 | 0.71 | 0.65 | 0.78 | 80 | -0.02 |
| 38 | Possibly attackable | 3.02 | 0.81 | 0.71 | 0.92 | 48 | 0.19 |
| 39 | Definitely doubly attackable | 2.67 | 0.66 | 0.63 | 0.80 | 92 | -0.11 |
| 40 | Definitely doubly attackable somehow | 2.66 | 0.69 | 0.67 | 0.80 | 88 | 0.14 |
| 41 | Possibly doubly attackable | 2.71 | 0.75 | 0.73 | 0.83 | 71 | -0.26 |
| 42 | Definitely attackable by pawns | 3.54 | 0.92 | 0.92 | 2.38 | 25 | 0.13 |
| 43 | Possibly attackable by pawns | 3.11 | 0.78 | 0.78 | 0.95 | 58 | -0.10 |
| 44 | Definitely attackable by knights | 2.91 | 0.72 | 0.71 | 0.84 | 77 | 0.24 |
| 45 | Definitely attackable by bishops | 2.60 | 0.64 | 0.61 | 0.80 | 95 | 0.15 |
| 46 | Possibly attackable by bishops | 2.60 | 0.68 | 0.64 | 0.85 | 89 | -0.07 |
| 47 | Definitely attackable by rooks | 2.63 | 0.65 | 0.64 | 0.75 | 93 | 0.07 |
| 48 | Possibly attackable by rooks | 2.74 | 0.70 | 0.69 | 0.77 | 81 | 0.00 |
| 49 | Possibly attackable without king | 2.72 | 0.70 | 0.63 | 0.79 | 82 | 0.19 |
| 50 | Possibly attackable without pawns | 2.63 | 0.67 | 0.62 | 0.73 | 90 | 0.17 |
| 51 | Definitely attackable by opponent | 3.25 | 0.87 | 0.91 | 0.77 | 31 | -0.03 |

Table 6: Synopsis feature descriptions, saliency estimates, and ablation study results (continued).

| # | Description | Loss PBS | Action PbS | Action PbS 0 | Action PbS 1 | Action PbS # | Ablation % Acc. Diff. |
|---|---|---|---|---|---|---|---|
| 52 | Possibly attackable by opponent | 3.15 | 0.84 | 0.90 | 0.81 | 37 | 0.06 |
| 53 | Definitely doubly attackable by opp. | 2.56 | 0.65 | 0.66 | 0.56 | 94 | -0.01 |
| 54 | Possibly doubly attackable by opp. | 2.67 | 0.71 | 0.73 | 0.66 | 79 | -0.13 |
| 55 | Definitely attackable by opp. pawns | 3.10 | 0.87 | 0.87 | 1.69 | 30 | 0.12 |
| 56 | Possibly attackable by opp. pawns | 2.84 | 0.77 | 0.77 | 1.10 | 62 | 0.30 |
| 57 | Definitely attackable by opp. knights | 2.78 | 0.69 | 0.70 | 0.59 | 87 | 0.21 |
| 58 | Possibly attackable by opp. knights | 2.14 | 0.52 | 0.51 | 0.55 | 102 | 0.08 |
| 59 | Definitely attackable by opp. bishops | 2.66 | 0.67 | 0.67 | 0.63 | 91 | 0.09 |
| 60 | Possibly attackable by opp. bishops | 2.16 | 0.53 | 0.52 | 0.56 | 101 | -0.09 |
| 61 | Definitely attackable by opp. rooks | 2.77 | 0.70 | 0.72 | 0.48 | 83 | -0.19 |
| 62 | Possibly attackable by opp. rooks | 2.10 | 0.51 | 0.53 | 0.47 | 103 | 0.20 |
| 63 | Possibly attackable by opp. w/o king | 2.55 | 0.64 | 0.63 | 0.64 | 96 | -0.17 |
| 64 | Possibly attackable by opp. w/o pawns | 2.45 | 0.62 | 0.61 | 0.62 | 97 | -0.13 |
| 65 | Possibly safe opposing king | 6.04 | 2.06 | 2.01 | 3.38 | 2 | 0.07 |
| 66 | Squares the opponent may move to | 2.40 | 0.60 | 0.60 | 0.60 | 98 | 0.01 |
| 67 | Possible castle state for opponent | 3.09 | 0.79 | 0.79 | 0.72 | 53 | 0.00 |
| 68 | Known squares | 4.94 | 1.52 | 1.67 | 1.45 | 6 | 0.13 |
| 69 | Own king's king-neighbors | 3.10 | 0.78 | 0.77 | 0.93 | 56 | 0.14 |
| 70 | Own king's knight-neighbors | 2.82 | 0.71 | 0.70 | 0.91 | 78 | 0.31 |
| 71 | Definitely opp. knights near king | 3.09 | 0.79 | 0.79 | 1.64 | 54 | 0.13 |
| 72 | Possibly opp. knights near king | 5.13 | 1.72 | 1.72 | 2.77 | 4 | -0.01 |
| 73 | Own king's bishop-neighbors | 2.74 | 0.69 | 0.68 | 0.86 | 85 | -0.10 |
| 74 | Definitely opp. bishops near king | 3.04 | 0.79 | 0.79 | 0.89 | 55 | 0.23 |
| 75 | Possibly opp. bishops near king | 5.23 | 1.75 | 1.75 | 2.41 | 3 | -0.11 |
| 76 | Own king's rook-neighbors | 2.76 | 0.69 | 0.68 | 0.83 | 86 | -0.13 |
| 77 | Definitely opp. rooks near king | 3.10 | 0.81 | 0.81 | 0.87 | 49 | 0.31 |
| 78 | Possibly opp. rooks near king | 4.45 | 1.40 | 1.40 | 1.55 | 10 | 0.05 |
| 79 | All own pieces | 5.26 | 1.36 | 1.09 | 2.47 | 11 | -0.01 |
| 80 | Definitely empty squares | 3.69 | 0.96 | 1.05 | 0.84 | 20 | -0.13 |
| 81 | May castle eventually | 3.11 | 0.81 | 0.81 | 1.26 | 46 | 0.24 |
| 82 | Possibly may castle | 3.05 | 0.77 | 0.77 | 0.63 | 68 | 0.05 |
| 83 | Definitely may castle | 3.04 | 0.77 | 0.77 | 0.87 | 66 | 0.12 |
| 84 | Own queens' rook-neighbors | 2.20 | 0.54 | 0.53 | 0.63 | 100 | 0.04 |
| 85 | Own queens' bishop-neighbors | 2.33 | 0.57 | 0.57 | 0.67 | 99 | 0.06 |
| 86 | Previous definitely not opp. pieces | 3.82 | 0.88 | 0.87 | 0.89 | 28 | -0.30 |
| 87 | Previous definitely opp. pawns | 4.16 | 1.16 | 1.18 | 0.88 | 12 | 0.14 |
| 88 | Previous definitely opp. knights | 3.02 | 0.77 | 0.77 | 0.72 | 63 | 0.12 |
| 89 | Previous definitely opp. bishops | 2.92 | 0.73 | 0.73 | 0.70 | 75 | -0.02 |
| 90 | Previous definitely opp. rooks | 3.60 | 1.01 | 1.02 | 0.56 | 16 | 0.05 |
| 91 | Previous definitely opp. queens | 3.93 | 1.11 | 1.11 | 1.00 | 15 | 0.22 |
| 92 | Previous definitely opp. king | 3.33 | 0.83 | 0.82 | 1.58 | 38 | -0.04 |
| 93 | Previous possibly not opp. pieces | 4.40 | 1.43 | 1.21 | 1.47 | 9 | -0.05 |
| 94 | Previous possibly opp. pawns | 3.27 | 0.83 | 0.82 | 0.92 | 39 | 0.21 |
| 95 | Previous possibly opp. knights | 3.04 | 0.78 | 0.78 | 0.73 | 60 | 0.22 |
| 96 | Previous possibly opp. bishops | 3.10 | 0.74 | 0.74 | 0.78 | 73 | 0.16 |
| 97 | Previous possibly opp. rooks | 2.94 | 0.77 | 0.79 | 0.45 | 64 | 0.10 |
| 98 | Previous possibly opp. queens | 3.02 | 0.74 | 0.74 | 0.81 | 74 | -0.07 |
| 99 | Previous possibly opp. king | 3.14 | 0.83 | 0.82 | 1.15 | 41 | 0.04 |
| 100 | Previous last from | 2.85 | 0.75 | 0.74 | 0.85 | 72 | -0.04 |
| 101 | Previous last to | 3.36 | 1.00 | 1.00 | 1.50 | 18 | 0.21 |
| 102 | Previous own capture | 3.05 | 0.84 | 0.84 | 1.13 | 36 | -0.09 |
| 103 | Previous opposing capture | 2.93 | 0.77 | 0.77 | 1.05 | 65 | 0.05 |