# OpenReview forum: "Deep Synoptic Monte-Carlo Planning in Reconnaissance Blind Chess"
_NeurIPS.cc/2021/Conference — NeurIPS 2021 Poster_

### Official Review · Reviewer_wWRf · 2021-07-16

**Rating:** 5
**Confidence:** 5

**Summary:**

The paper describes Penumbra, a top reconnaissance blind chess program, with focus on the state abstraction and the Monte Carlo planning algorithm.

**Ethical Concerns:**

-

**Limitations And Societal Impact:**

Yes

**Main Review:**

The main algorithmic contribution is probably the deep synoptic Monte Carlo planning algorithm, but there is little evaluation of it, other than the game program being strong (which can be due to other factors as well). It would be useful to have a more in-depth comparison with other planning algorithms used by the top competitors.

The paper includes a detailed description of the network architecture and other implementation details, but I am not sure that some of these details are too important for the planning algorithm.

Reconnaissance blind chess is a complex domain with many interesting properties for the NIPS audience. Therefore, I regard it as a very good benchmark. However, the particular algorithmic ideas need to be properly evaluated and compared directly with possible alternatives, and not just indirectly through the performance of a game program that happens to include it. As it is, the paper seems a solid paper for a game conference, but I am not sure that it presents sufficient interest for the NIPS audience.

**Time Spent Reviewing:**

5

---

> ### Author Response · Authors · 2021-08-10
> **Response to Reviewer wWRf**
>
> ### Re: Little evaluation
> The results presented in Tables 2 through 4 required approximately 700 GPU days to produce, given that the games are played at standard time control where each side gets 15 minutes per game.  Retraining the model with each feature removed required approximately 60 GPU days.  Even Figure 6 (a) required a non-trivial amount of computation.  Only a single computer with four GPUs was available for this research.
>
> Are tighter confidence intervals or bigger tables of Elo scores really necessary?  What "other factors" are you referring to?
>
> ### Re: In-depth comparison
> The paper already devotes more than a full page to describing related work and the algorithms used by competitors.  Consider reading the cited papers if you're interested in a more in-depth comparison.  Devoting more space to other algorithms in this paper does not seem appropriate.  Do you have any specific questions about how algorithms compare?
>
> ### Re: Detailed description
> While some details may not be too important for the planning algorithm, including details makes the paper more reproducible.  Are there any specific details that seem worth removing from the paper?
>
> ### Re: Algorithm evaluation
> There is literally no other way to empirically evaluate an algorithm for playing games other than through the performance of a program that includes the algorithm.  Note that naive implementations of "possible alternatives" are intractable in RBC.
>
> The paper already compares directly with four other algorithms, and it includes the PenumbraSimple ablation which shows that the program's static board analysis does not contribute significantly.
>
> ### Re: Audience
> Note that NeurIPS is **the** conference that hosts RBC competitions.

---

### Official Review · Reviewer_qSQf · 2021-07-16

**Rating:** 7
**Confidence:** 4

**Summary:**

This paper describes a Reconnaissance blind chess (RBC) program Penumbra that won the 2020 RBC rating competition. Penumbra’s method is called “deep synoptic Monte Carlo planning” which is a set of bandit, belief state summarization, imitation learning techniques that together act on an underlying POMDP model.


**Limitations And Societal Impact:**

The authors could discuss further limitations given some of the questions posed above. They do point out that they are not trying to compute a Nash equilibrium.

**Main Review:**

Originality: This combination of techniques that leads to an effective algorithm for playing RBC, in my opinion, is novel and interesting. The most original part of the methodology seems to be the synoptic function (summary function) that takes as input approximate infostates and outputs a fixed size summary through the use of logical operators that “scalarize” across the set of possible world states (= a belief state).

Quality:
The approach to summarizing approximate infostates is clever and makes sense. However, as a reader, it would be nice to see a clearer discussion regarding the related work with respect to the synopsis function.
Can the authors discuss what happens when there is no training data for an opponent? How well does Penumbra handle playing against a new opponent? Is there a possibility of catastrophic outcomes if Penumbra uses the model of one opponent but plays against another? These would be useful to discuss in a limitations section.


Clarity: The paper is generally well-written with intuitive illustrations. It was a pleasure to read. I have a few points below --- it would help to make some notation more precise or give explanations when there is possible ambiguity.
Line 106: Clarify the statement “equivalently, the set of all possible action histories from p’s perspective”.
Line 104: Why is the action a function defined on a subset of world states?
The discussion about n_particles and the unweighted particle filter deserves more clarity -- I was a bit lost in reading this part as few details are given.
Typo in Algorithm 3: Bandi -> Bandit
How are SoonWin, SoonLose, and PieceCount used?

Significance: Given the performance of Penumbra in the RBC competition, my judgment is that this constitutes a significant contribution to gameplay AI. The method of summarizing infostates seems generalizable to other settings and perhaps could be studied theoretically: how much are we losing by collapsing infostates in this manner? Could the authors give a discussion on this point?

**Time Spent Reviewing:**

1.5

---

> ### Author Response · Authors · 2021-08-10
> **Response to Reviewer qSQf**
>
> Thank you for the valuable feedback!
>
> ### Re: Unrecognized and new opponents
> When there is no training data for an opponent (such as a new opponent), the policy head from the `All` headset is used to model the opponent's actions.  Clarification will be added.
>
> Table 2 includes results that show what happens when Penumbra uses the model of one opponent but plays against another (for a few incorrect pairings, at least).  Additional games have been played against misrecognized Trout since submitting the paper.
>
> Yes, a wrong model of the opponent could possibly lead to catastrophic outcomes (such as losing almost all games).  This will be added to the discussion section.
>
> ### Re: Possible action histories
> The statement about "possible action histories" is meant to point out the equivalence of the given definition of an information state with the (standard) definition of an information set as in Osborne & Rubinstein.  That citation will be restored.  An early draft expounded on action histories, but that part was removed as unnecessary.  Mentioning action histories may be removed entirely if you'd prefer.
>
> ### Re: Action functions
> Actions are defined as functions because it seemed natural to do so.  Each action is only defined on a subset of world states because not all actions may be taken from all world states.  For example, an action $a \in \mathcal{A}$ might be "move a rook from e2 to e4", and that action is only possible in world states that have a friendly rook on e2.
>
> ### Re: Particle filter
> A sentence will be added to clarify that the particles in the particle filter are approximate infostates from the opponent's perspective.
>
> (Alternatively, the sampling algorithm could be thought of as block sequential Monte Carlo or even Gibbs sampling where the samples are action histories and each action component is conditioned on the previous actions.)
>
> ### Re: Extra output heads
> SoonWin, SoonLose, and PieceCount are included in training for regularization.  They are not used while playing the game.  (A new ablation study could assess whether or not including these heads actually helps.)
>
> ### Re: Effect of collapsing infostates
> Indeed, it's not clear how much we are losing by collapsing infostates in this manner, and this question deserves further investigation.
>
> ### A note on notation and citations
> A set of possible world states is equivalent to an information state (AKA information set) in environments without private actions.  The ISMCTS paper (Cowling, 2012) even defines information set in this way.  However, in some settings such as RBC, the set of possible world states contains less information than the set of possible action histories (which is how Osborne & Rubinstein define an information set).
>
> Russell & Norvig equates information sets with belief states.  However, belief states may incorporate prior information (about another agent for example) and assign probabilities to world states, while information sets only include objective information about possible histories.  So, the two concepts are not equivalent, and it's important to be careful with these terms.
>
> (Oops, two of these citations were accidentally missing from the paper's references, and the third was accidentally cited from 2010 instead of 2020.  The references will be updated.)

---

> > ### Comment · Reviewer_qSQf · 2021-08-30
> > **After response**
> >
> > Thanks for the response! I plan to keep my score. I hope the clarifications can be addressed in a revised version.

---

### Official Review · Reviewer_Yimf · 2021-07-16

**Rating:** 6
**Confidence:** 3

**Summary:**

This paper presents Deep Synoptic Monte Carlo Planning, the technique behind the 2020 champion Reconnaissance Blind Chess (RBC) program Penumbra.  This method samples possible world states to approximate information sets, which it then abstracts uses synopses.  These abstracted information sets are then used monte-carlo planning to determine actions to take in the game.  The approach, with some RBC specific tweaks, is evaluated in the RBC domain and shown to be very successful.


**Limitations And Societal Impact:**

Discussion was adequate

**Main Review:**

The core idea of abstracting information sets using the stochastic synopses is, to my knowledge, a novel one and appears to be very successful.  The problem is well-motivated and the results significant for this challenging problem.  On the other hand, I found the paper fairly dense, and the individual contributions not granted enough space for their explanations to be clear.  The notation was not clearly introduced, especially as it related to individual parameters.  In my opinion, this is important work, but the current presentation could be significantly improved to make the contributions more clear and understandable.

Comments
- I found the algorithm references to be slightly confusing, as the algorithms aren't named, (except for Alg. 1, 2, etc) but then are referred to in other algorithms.  I was able to figure out what was going on, but this made the description a bit harder to follow.
- Many of the parameters and variables that are used to define the algorithms are not described (that I could find), anywhere in the text or algorithm descriptions.  For example, an $m$ value is referred to multiple times in the text (page 7 line 193, page 8 line 204-205) but the only place it appears is in Algorithm 1.  I would have appreciated some discussion in the text of what this $m$ value controls or influences in the operation of the algorithm.  It was unwieldy to have to flip back from page 8 to the definition of Alg. 1 and then try to figure out how the versions of Penumbra that were evaluated differ.
- In a similar fashion I could see no discussion or explanation of $b$, $d$, $n_{vl}$, and $\nu$, which are referred to in Alg. 3.
- Towards the end of Alg 3. "Bandit" is mispelled "Bandi" and the final parameter passed to it is a 0, where it should be false to match other usages.
- I found the explication of the Synopsis method, a key contribution of the work, hard to follow. In particular, I failed to see the connection between formula 1 and the contents of Figure 4.  Specifically, the example following formula 1 leads me to see each $x_j$ as a different world states possible (in the abstraction set), while the $g_i$ function used refers to moving to a specific square on the board.  Combined with the AND function, this gives a single binary value that tells whether or not it is guaranteed that the knight can move to that square.  In Figure 4, we see bitboards, the same size as the chessboard, which are described as summarizing things like all the places an enemy pawn could appear. The connection between these two was not immediate to me.  Would the bitboard for the example feature (knight movement) have a black bit everywhere that the knight could definitely move, and a white bit where it might not be able to move?
- How were the 14 different headsets of the network used in the planning?  This was an interesting detail to me that I would have loved to have more of an explanation of.  The text merely says that the policy heads and the All value head were used for planning, but how this occurs was left undescribed.
- Why does having a smaller l for the opponent lead to more cautious play?  My first thought is that the fewer states that are considered in each opponent infoset the more chance there is that we miss the truth and thus it would seem less cautious and more risky to have fewer.  I would have loved more discussion about this.
- What did it mean for an opponent to be "recognized as" vs. "unrecognized" in the experimental results? I couldn't find a discussion of what this meant or entailed in the paper anywhere.
- In the discussion of Saliency (Section 7) I was expecting some conclusions about which type of synopsis features were found to be most impactful.  Am I correct in assuming that the features shown in Figure 6 were also found to be salient in other training runs?
- As far as the saliency features, how were these determined?  Were these hand-designed, or learned somehow?

In summary, I really like this work, the problem it addresses, and the methods it uses.  But I was left with many questions and confusions as I tried to parse the contributions and what had actually been done from the paper itself.

Addendum: I appreciated the author's responses and am more confident that with the changes they describe the work will be clear enough.

**Time Spent Reviewing:**

3

---

> ### Author Response · Authors · 2021-08-10
> **Response to Reviewer Yimf**
>
> Thank you for the valuable feedback!
>
> ### Re: Synopsis method
> Yes, your understanding of Formula 1 and Figure 4 is correct, and a description of the connection between them will be added.
>
> ### Re: "Recognized" opponents
> An opponent is "recognized" if and only if the opponent is identified as the sole-source of training data for one of the headsets.  That will be added.
>
> ### Re: Headsets in planning
> While playing against an opponent that is "recognized", the policy head ($\hat{\tau}$) of the corresponding headset is used while sampling the opponent's moves for progressing the particle filter (Alg. 2) and while sampling the opponent's moves for constructing the UCT tree (Alg. 3).  That will be added!
>
> ### Re: Cautious play
> Having smaller $\ell$ for the opponent leads to more "cautious" play since each approximate infostate is guaranteed to contain the correct ground truth in the playout.  See Alg. 3: $I \gets \text{ random } \ell \text{ states in } I \underline{\text{ including } a_t(x_t)}$.
>
> So, when $\ell$ is small for the opponent, the opponent has higher-quality information in the constructed UCT tree.  Thus, the opponent is more likely to notice and counter a sneak-attack in (the imagined future of) the UCT tree.  So, the UCT tree search is more likely to discourage risky sneak attacks since the opponent is more likely to notice and respond effectively in the tree search.  That's the idea, anyway.  Does that make sense?
>
> An explanation will be added.
>
> ### Re: Features and saliency
> The synopsis features were hand-designed.  Many of them are straightforward and natural given the rules of the underlying game of chess. Some of them are near duplicates of each other.  Learning a useful and compact representation of an RBC infostate is an interesting future research direction.
>
> The saliency statistics were designed to be similar to standard saliency measurements but averaged over examples and over the training steps in order to accommodate the discrete input space.
>
> Table 5 and 6 in the appendix contain the saliency statistics for each feature averaged over 5 runs.  Figure 6 (c)-(f) shows results from one training run only because that was easier to plot.  Plotting the average should be possible too.  Yes, the most-salient features were consistent between training runs.
>
> ### Re: Notation
> Section 4 will be updated to introduce the notation more clearly, and Table 1 in the appendix could be moved into the main paper.  The algorithm names will be made consistent between the algorithm bodies and titles.
>
> Indeed, the variable $m$ was only defined by its usage in Algorithm 1 and given a brief description in Table 1 in the appendix.  A description of $m$ as the control over how policies are mixed in the bandit algorithm will be added.
>
> The variables $b$, $d$, and $n_{vl}$ are briefly described in Table 1 in the appendix, and $\nu$ is a value function. Yes, descriptions should and will be added to the paper.
>
> Good catches, thank you for pointing all of this out!  Please feel free to share any additional concerns or questions.

---

### Official Review · Reviewer_94iT · 2021-07-22

**Rating:** 7
**Confidence:** 3

**Summary:**

The paper describes an approach to planning in imperfect information games called DSMCP. The algorithm is the basis for a system that won the 2020 reconnaissance blind chess competition. The algorithm views the game as a POMDP by assuming a fixed policy for the opponent. Infostates are represented compactly using a particle filter where each particle represents a set of world states with a maximum size. These are further compressed as "synopses", which are conjunctions or disjunctions of Boolean features over all the states in the set. The synposes are then input to a neural network that has been trained to imitate one of a set of possible policies including several other RBC programs. Policy action choices are mixed with UCB action choices to create a search tree. Saliency of different synopses is found to be somewhat correlated with decreased performance when removing that feature.

**Limitations And Societal Impact:**

I see little potential for negative societal impact of this work

**Main Review:**

## Originality

The builds on existing tree search algorithms for imperfect-information games, and algorithms that combine tree search and trained policies. The main original contribution seems to be the formulation of the synopses. There are also numerous design choices made to make the problem tractable that, taken collectively, result in an original system. Related work is cited extensively.

## Quality

The techniques and specific design choices employed are mainly heuristics, but they are well-motivated by the goal of tractability. The key experimental results are the Elo scores from the RBC tournament, which demonstrate that the method achieves a higher playing strength than its competitors. There is a lack of ablation-type results that drill down into the impact of specific choices -- e.g., selection of particular synopsis features, choice of set sizes for approximations, etc. (at least in the main paper). Some results are also presented with little in the way of interpretation -- specifically, Section 7 and Figure 6. What do the results on saliency tell us about the synopsis features, or how could they guide the selection of features?

## Clarity

The paper is generally well-written but is extremely information-dense. It feels like there was a struggle to fit in all the content, and the presentation is fairly terse.

## Significance

Solution methods for imperfect information games are of broad interest. This work does not make any ground-breaking advances, but demonstrates a number of practical techniques for making larger problems tractable. Since the agent won an organized RBC competition, presumably the approach will have some impact on the design of other RBC agents.

## Notes:

Alg. 3: What is $\nu$?

Alg. 4: Should ChooseAction take (N, Q) as arguments?

159-160: Is it correct that $|X| = F$ (the size of the state set is the same as the number of features)? Should it be $|X| = \ell$?

**Time Spent Reviewing:**

2.5

---

> ### Author Response · Authors · 2021-08-10
> **Response to Reviewer 94iT**
>
> Thank you for the valuable feedback!
>
> ### Re: Ablations
> PenumbraSimple and PenumbraNetwork are ablation-type results that approximate the contribution of the static board analysis and the tree search, respectively.  Additional ablations that require playing games to estimate strength are expensive, but running another one is still possible if you're interested in a particular ablation study.
>
> ### Re: Saliency interpretation
> Indeed, saliency was somewhat correlated with decreased predictive performance when removing a feature.  While the noise in the training process makes it difficult to draw certain conclusions, features with high saliency are likely to be the most important.  Also note that several features are similar to others, and so removing one is likely to increase reliance on other similar features.  The saliency results suggest features for removal and for expansion as well as guide other researchers while creating new RBC programs.
>
> ### Re: Terse presentation
> Yes, fitting the content into the nine pages was tricky.  The additional page in the camera-ready version will admit more clarifying commentary.
>
> ### Re: Notes
> In Algorithm 3, $\nu$ is a value function, such as the output of a value head of the neural network.  The definition will be added.  Yes, ChooseAction should take $\textbf{N}$ and $\textbf{Q}$ as arguments.
>
> Good catch, $F$ should not be used as the subscript to the last element in $X$ in equation 1 and on line 160. $\ell$ is the upper bound on the number of world states to track at a time. That is, for an approximate infostate $X$, $|X| \le \ell$. And, $F$ is the number of binary synopsis features.  The two values are indeed distinct; $\ell = 128$ and $F = 8 * 8 * 104 = 6656$ in this paper.
>
> Thank you for pointing these out!

---

### Decision · Program_Chairs · 2021-09-27

**Decision:**

Accept (Poster)

**Comment:**

The reviewers had several concerns about this work, but after discussion these were generally agreed to be fixable issues - with one exception. One reviewer felt very strongly that RBC would not be sufficient to demonstrate the generality of the contribution. I suspect that some of the disagreement here may be an issue of different sub communities having different expectations. While RBC may be a familiar and accepted benchmark in one sub community, it may not be familiar in others - even some that aren't that far away.

The potential narrowness of the appeal of this paper indicates that a poster may be the more appropriate route.

I would also encourage the authors to find some way to make their contribution appeal more broadly if is is practical to (convincingly) make the case that it has broader application than to a single benchmark.